# Gut Microbiota Characteristics of People with Obesity by Meta-Analysis of Existing Datasets

**DOI:** 10.3390/nu14142993

**Published:** 2022-07-21

**Authors:** Jinhua Gong, Yun Shen, Hongcheng Zhang, Man Cao, Muyun Guo, Jianquan He, Bangzhou Zhang, Chuanxing Xiao

**Affiliations:** 1School of Medicine, Xiamen University, Xiamen 361102, China; gongjinhua2068@163.com; 2Sport Hospital Attached of ChengDu Sport University, Chengdu 610041, China; shenyun0827@hotmail.com; 3Xiamen Treatgut Biotechnology Co., Ltd., Xiamen 361101, China; hcmrzhang@163.com (H.Z.); caoman@treatgut.com (M.C.); 4Pulmonary and Critical Care Medicine, Anyang District Hospital, Anyang 456000, China; iguomuyun@163.com; 5College of Rehabilitation Medicine, Fujian University of Traditional Chinese Medicine, Fuzhou 350122, China; hejianquan08@163.com; 6Department of Rehabilitation, Zhongshan Hospital of Xiamen University, School of Medicine, Xiamen University, Xiamen 361102, China; 7School of Pharmacy, Fujian University of Traditional Chinese Medicine, Fuzhou 350122, China; 8Department of Gastroenterology, The Second Affiliated Hospital of Fujian University of Traditional Chinese Medicine, Fuzhou 350003, China

**Keywords:** gut microbiota, obesity, 16S rRNA, BMI, humans, meta-analysis

## Abstract

Obesity is a complex chronic, relapsing, progressive disease. Association studies have linked microbiome alterations with obesity and overweight. However, the results are not always consistent. An integrated analysis of 4282 fecal samples (2236 control (normal weight) group, 1152 overweight, and 894 simple obesity) was performed to identify obesity-associated microbial markers. Based on a random effects model and a fixed effects model, we calculated the odds ratios of the metrics, including bacterial alpha-diversity, beta-diversity, Bacteroidetes/Firmicutes ratio, common genera, and common pathways, between the simple obesity and control groups as well as the overweight and control groups. The random forest model was trained based on a single dataset at the genus level. Feature selection based on feature importance ranked by mean decrease accuracy and leave-one-out cross-validation was conducted to improve the predictive performance of the models. Chao1 and evenness possessed significant ORs higher than 1.0 between the obesity and control groups. Significant bacterial community differences were observed between the simple obesity and the control. The ratio of Bacteroidetes/Firmicutes was significantly higher in simple obesity patients. The relative abundance of *Lachnoclostridium* and *Faecalitalea* were higher in people with simple obesity, while 23 genera, including *Christensenellaceae_R-7_group*, *Akkermansia, Alistipes*, and *Butyricimonas*, etc., were significantly lower. The random forest model achieved a high accuracy (AUC = 0.83). The adenine and adenosine salvage pathway (PWY-6609) and the L-histidine degradation I pathway (HISDEG-PWY) were clustered in obese patients, while amino acid biosynthesis and degradation pathways (HISDEG-PWY, DAPLYSINESYN-PWY) were decreased. This study identified obesity microbial biomarkers, providing fertile targets for the management of obesity.

## 1. Introduction

Obesity is a complex, chronic, relapsing, progressive disease in which abnormal or excess body fat accumulates in adipose tissue. WHO criteria define overweight in adults as a BMI of 25.0–29.9 kg/m^2^ and obesity as a BMI of 30.0 kg/m^2^ or higher [1,2]. Obesity affected 670 million adults worldwide in 2016 [3]. The World Obesity Atlas 2022 predicted that 1 in 5 women and 1 in 7 men will be living with obesity by 2030, equating to over 1 billion people globally [4]. Overweight and obesity are risk factors for major noncommunicable diseases, including type 2 diabetes, cardiovascular disease, and cancer [5]. Obesity incurs the burden of major obesity-related noncommunicable diseases and reduces lifespan. Moreover, obesity is the second leading predictor after age for COVID-19 complications and mortality [6]. The WHO set a global action plan for the prevention and control of noncommunicable diseases from 2013–2020, including halting the rise in diabetes and obesity by 2025 [7]. The pathogenesis of obesity includes genetic susceptibility, biology, health care access, mental health, sociocultural factors, nutritional transitions, economics, commercial determinants, and environmental determinants [4,5]. The gut microbiome impacts body fat by regulating host metabolism, such as bile acids, indole propionic acid, branched chain amino acids, and endocannabinoids [8].

Reduction in sequencing costs and advances in bioinformatics have made it easier to capture more views of the association between the gut microbiota and obesity [9]. Transplantation of normal microbiota harvested from conventionally raised animals to adult germ-free mice produces a 60% increase in body fat content despite reduced food intake [10]. Transplantation of fecal microbiota from adult female twin pairs discordant for obesity into germ-free mice fed low-fat mouse chow can increase total body and fat mass [11]. Although most studies have shown higher abundances of Firmicutes in obese people, a few studies have shown different results [12,13,14,15]. Discrepant findings may be due to differences in research methods, clinical characteristics, or study heterogeneity (geography, ethnicity, diet). More individual patient-level meta-analyses—preferably with standardized bioinformatics pipelines—are required to explain the heterogeneity given that few of the previous systematic reviews [15,16,17] were quantitative systematic reviews. Recently, one meta-analysis including 32 studies was included in the qualitative synthesis, and 11 studies were included in the quantitative synthesis. Pinart, M., et al. investigated microbial diversity and richness, differences in the relative abundance of bacteria at the phylum level, and significant differences in the relative abundance of bacteria at the genus level between obese and nonobese persons [15]. The limitation is that the analysis was unable to provide species-level information. Almost at the same time, a large cohort was employed to analyze the abundance of 50 prevalent and well-characterized gut microbes using a validated quantitative PCR method [18]. The limitation of this study is that analysis by bioinformatics methods to identify specific genes and pathways was not performed.

In this study, we performed a meta-analysis using fecal 16S rRNA gene sequence data from eight studies. By analyzing all datasets in a uniform manner, we aimed to (i) clarify the differences in fecal bacterial diversity and communities in patients with obesity, (ii) identify a universal set of microbial markers to predict obesity, and (iii) predict the functional pathways of microbial communities.

## 2. Materials and Methods

### 2.1. Database Search and Study Selection

Based on the Preferred Reporting Items for Systematic Reviews and Meta-Analyses (PRISMA) standard (Figure 1), the following keywords were selected to search the literature that were included in the PUBMED, EMBASE, and COCHRANE databases before August 2021: obesity and gut microbiota and human and 16S rRNA. Body mass index (BMI) was calculated as weight in kilograms divided by the square of height in meters. Obesity was identified and classified by BMI according to WHO definitions [19]: normal weight as a reference (BMI ≥ 18.5 but <25 kg/m^2^), overweight (BMI ≥ 25 but <30 kg/m^2^), and obesity (BMI ≥ 30 kg/m^2^). The criteria for study inclusion were as follows: (1) studies were based on human fecal samples from obese, overweight, and healthy normal weight subjects who had not taken antibiotics for six months; (2) samples were sequenced by NGS for the 16S rRNA gene; (3) raw sequencing data, barcodes, and metadata were publicly available or provided by the authors until 31 August 2021, upon request by email; and (4) the type of obesity was simple obesity without complications and comorbidities. Ultimately, a total of eight sequencing datasets and metadata [13,20,21,22,23,24,25] were included in the downstream meta-analysis (Table 1). After finishing quality filtering, we obtained a total of 4282 samples (2236 control (normal weight) group, 1152 overweight, and 894 simple obesity) for downstream analyses (Table 1).

### 2.2. Microbiome Data Processing

The most frequently sequenced fragment with the Illumina (MiSeq or HiSeq) or Ion Torrent platform (PGM or S5) among the included studies was the V4 or V3-V4 region of the 16S rRNA gene (Table 1). Due to the different sequencing platforms and hypervariable regions of the 16S rRNA gene, we analyzed the different cohorts in a uniform analytical pipeline to minimize the impact of these differences. Briefly, raw sequence data were preprocessed. Clean reads with high quality were obtained through sequence merging and quality control. First, when Fast Length Adjustment of Short Reads(FLASH) (V1.2.11) [26] was used to assemble paired-end reads for the V4 region, the -x 0.15 option was selected to control the maximum mismatched base pairs ratio in the overlap area, and the -M 150 or -M 250 option was selected to control the maximum length of the overlap area. Then, cutadapt (V1.13) [27] was used to trim and filter the sequence data processed by FLASH, including removing adapter sequences and discarding sequences with fewer than the specified number of bases. Subsequently, sequences were quality filtered by Usearch with the -fastq_maxee 1.0 option. After quality control, unique sequences were obtained by eliminating redundancy, and they were sorted in descending order according to sequence abundance. Meanwhile, singletons in the sequence data were removed. Clean reads were clustered into operational taxonomic units (OTUs, similarity threshold 97%) by using Usearch with the UPARSE-OTU algorithm [28]. Chimaera detection was performed using UCHIME [29] against the Ribosomal Database Project (RDP) classifier [30]. For all taxonomic and diversity analyses, samples with sequencing depths less than 10,000 sequences in the OTU table were removed. The OTU table was rarefied to the lowest sequencing depth within each study. The pathway abundances in each dataset were predicted by PICRUSt2 [31].

### 2.3. Statistical Analysis

Based on the OTU tables derived from each study, alpha-diversity indices between the simple obesity and control groups as well as the overweight and control groups were calculated, including bacterial richness (observed OTUs, Chao1, ACE), Shannon index, Simpson index, and evenness (J). Significance tests of alpha-diversity indices were conducted by the Wilcoxon test method. Principal coordinates analysis (PCoA) based on Bray–Curtis distance at the genus level was utilized for beta-diversity to visualize the differences in microbial community structure across samples. Significance tests of beta-diversity indices were determined using permutational multivariate analysis of variance (PERMANOVA) with 10^4^ permutations in vegan [32]. Meta-analysis of bacterial alpha-diversity indices and microbial taxa among the eight studies was performed in the metafor package [33] to discover the consistency using both the random effects (RE) model and the fixed effects (FE) model. PICRUSt2 was utilized for gut microbiota functional prediction in each dataset [31]. We further identified pathways shared between the simple obesity and control groups and the overweight as well as control groups. Generally, we calculated the odds ratios (ORs) of these metrics by assigning any value above the median of the metric within the study as positive.

Random forest (R packages, caret [34]) models were trained for different cohorts, and datasets combined all studies together at the common genus levels to test whether a mixture of featured taxa can predict simple obesity or overweight. We evaluated their performance using leave-one-out (LOO) cross-validation and scored the predictive power in a receiver operating characteristic (ROC) analysis. In the first step, we ranked the common genera importance based on mean decrease accuracy. Next, we conducted stepwise feature selection with 10-fold cross-validation to avoid overfitting and overoptimistic evaluation. This method was employed to select predictive microbial features and eliminate uninformative features [35]. The area under the ROC curve (AUC) was calculated to evaluate the discriminatory power of genera.

## 3. Results

### 3.1. Microbiome Profile Differences among the Simple Obesity, Overweight, and Control Groups

There were significant differences in the overall microbial community structure among all groups when combined all samples from the eight studies together (PERMANOVA, F = 38.263, *p* < 0.001). However, the PCoA plot based on Bray–Curtis distance reflected that samples were clustered mainly by individual studies, which may be attributed to sample populations, DNA extraction methods, sequencing regions of the 16S rRNA gene, and sequencing platforms adopted by individual studies (Figure 2). To more objectively reflect the consistent differences of the gut bacterial community between the simple obesity and control groups, as well as between overweight and control groups, we performed a meta-analysis on the microbial metrics based on each individual study in the following analysis.

We evaluated the differences in alpha-diversity metrics between the simple obesity and control groups. Chao1 and evenness showed significant ORs higher than 1.0 (Figure 3A), indicating that these indices of the control group were significantly higher than those of the simple obesity group. Even when compared in individual studies, two of the eight studies observed significantly higher microbial richness in the control than in the simple obesity group (Appendix A). One study showed that Shannon and Simpson diversities were significantly higher in the control group than in the simple obesity group (Appendix A). Significantly higher evenness in the control group than in the simple obesity group was observed in two of the eight studies (Appendix A). Similarly, differences in alpha diversity between the overweight and control groups were compared. None of the alpha-diversity metrics were significantly different between the overweight and control groups (Appendix A) in the RE model. Two studies observed significantly higher microbial richness in the control than in the overweight group (Appendix A). One study showed that Shannon and Simpson diversities were significantly higher in the control group than in the overweight group (Appendix A). One study had significantly higher evenness in the control group than in the overweight group (Appendix A).

When evaluating differences in the entire bacterial community among the simple obesity, overweight, and control groups by PERMANOVA, significant differences in overall communities between obese and control individuals were obtained in four of the eight studies (Appendix A). There were significant differences between the overweight and control groups in two of the above five studies (Appendix A). By RE model, significant bacterial community differences were observed between the simple obesity and the control, but no significant differences between the overweight and control groups (Figure 3B and Appendix A). The ORs of the Bacteroidetes/Firmicutes ratio were significantly lower than 1.0 for both the simple obesity and overweight groups than the control group (Figure 3C and Appendix A). As for predicted bacterial pathways between the simple obesity and control groups, a total of 19 pathways possessed significant ORs higher than 1.0 for the control group. The total 19 pathways include L-lysine biosynthesis I (DAPLYSINESYN-PWY), aromatic biogenic amine degradation (bacteria) (PWY-7431), pyruvate fermentation to acetone (PWY-6588), guanosine nucleotides degradation III (PWY-6608), etc. For the simple obesity group (Figure 3D), the total five pathways had significant ORs lower than 1.0. The total five pathways include lactose and galactose degradation I (LACTOSECAT-PWY), adenine and adenosine salvage III (PWY-6609), superpathway of thiamin diphosphate biosynthesis II (PWY-6895), pyrimidine deoxyribonucleosides salvage (PWY-7199), and L-histidine degradation I (HISDEG-PWY). Between the overweight and control groups, there were six pathways with significant ORs lower than 1.0 for overweight patients, whereas none of the pathways possessed significant ORs higher than 1.0 for the control group (Appendix A).

For the purpose of further identifying the significantly different taxa among the control, overweight, and simple obesity groups, we calculated the ORs of all common taxa in each study. We identified 25 genera that were significantly associated with simple obesity (Figure 4A) and 14 genera that were significantly associated with overweight (Appendix A). *Lachnoclostridium* and *Faecalitalea* had significant ORs lower than 1.0 for the simple obesity group in the RE models. Twenty-three genera, including *Christensenellaceae_R-7_group*, *Akkermansia*, *Alistipes*, and *Butyricimonas*, etc., possessed significant ORs higher than 1.0 for individuals with normal weight (Figure 4A), which means that these bacteria were scarce in simple obesity patients. For overweight individuals, only one genus possessed significant ORs lower than 1.0, while 13 genera had significant ORs higher than 1.0 for individuals with normal weight (Appendix A). The heat map showed the fold change of the genera with significant ORs between the simple obesity and control groups in individual studies (Figure 4B). Most of the genera were enriched in the control group. In addition, we presented the relative abundance of genera in the control group with that in the simple obesity group (Figure 4C). The top five taxa in terms of relative abundance at different taxonomic levels in each study, including bacterial phyla, class, order, and family were presented in Appendix A. In most of the included studies, the changes in the relative abundance of Bacteroidetes and Firmicutes phyla were consistent with the ORs in the Bacteroidetes/Firmicutes ratio (Figure 3C). In the simple obesity group, the relative abundances of Bacteroidetes and Proteobacteria were higher, whereas Firmicutes was lower. To sum up, there were significant changes in the microbial community composition of patients with simple obesity.

### 3.2. Metagenomic Simple Obesity and Overweight Classification Models

Is there a specific set of microbes that can be used to identify or predict simple obesity? To obtain such biomarkers, we first constructed a random forest classifier model based on genera shared among the simple obesity, overweight, and control groups of all included studies. Tenfold cross-validation was carried out on the dataset of a single study, and modeling was based on common features in all studies. However, the model performance of the training model of a single study was not ideal. Hence, we took into account feature selection based on feature importance ranking (mean decrease accuracy) for training the model of individual studies. After feature selection based on 90 common genera between the simple obesity and control groups, the AUC values of the models were improved (Figure 5A). For example, the specificity and sensitivity of the PRJCA004023 simple obesity classifier model were 49.25% ± 0.262 and 48.75% ± 0.268 (AUC = 0.506), respectively, before feature selection. After feature selection, the specificity and sensitivity of the PRJCA004023 simple obesity classifier model were increased to 66.92% ± 0.251 and 81.66% ± 0.194 (AUC = 0.8099), respectively. In addition, five studies with overweight data were modeled based on 162 features at the genus level shared by the overweight and control groups. Similarly, according to the importance ranking of 162 features, overweight models were trained after feature selection (Appendix A).

We further assessed whether including data from all but one study in model training could improve prediction in the remaining hold-out study (LOOS validation). The LOOS performance of genus-level simple obesity models ranged from 0.70 to 0.83 (Figure 5B). The LOOS performance of overweight models ranged from 0.66 to 0.67 (Appendix A). These results suggest that the inclusion of multiple studies in the training set of a classifier did substantially improve its predictive performance relative to models trained on data from a single study. When pooling all studies together, the AUC value of the model (Total study) achieved 0.83 after feature selection (Figure 5B). Subsequently, a total of 38 features were retrieved with the ORs of each feature according to the random effect model (Figure 5C). Similarly, we extracted 48 features from the overweight model and ranked the importance of the features and the ORs (Appendix A). Furthermore, a total of eight genera were both selected as important features for the simple obesity model and significantly different between simple obesity and control (Table 2) and a total of four genera for overweight (Table 3).

## 4. Discussion

An integrated analysis of 4282 fecal samples (2236 control (normal weight) group, 1152 overweight and 894 simple obesity) was performed to identify obesity-associated microbial markers. Though observed, ACE, Shannon, Simpson were no significant differences, Chao 1, J and beta-diversity changes were statistically significant between the simple obesity and control groups and therefore should be used as general features differentiating normal and obese human gut microbiota across populations. One previous meta-analysis showed no significant differences in the Shannon index between obese and nonobese individuals in the meta-analysis [15]. This is consistent with our research. Although conclusions are inconsistent in different researches for Chao1, the meta-analysis does not include quantitative analysis for Chao1. Gordon and colleagues advocated that the ratio of Bacteroidetes/Firmicutes in people with simple obesity was lower [15], which might be considered as a biomarker of dysbiosis for simple obesity. In our study, compared with the control group, the abundance of Firmicutes was significantly lower in the simple obesity group, while Bacteroidetes was significantly higher, and the ratio of Bacteroidetes/Firmicutes was increased, which was consistent with the research results of Schwiertz A et al. [36].

The gut microbiota affects the energy balance of the host by regulating the genes related to fat absorption and storage [37]. Meanwhile, the dysbiosis of gut microbiota leads to the increase of endotoxin in the circulating system of the host, and induces chronic and low-level inflammation, leading to obesity and insulin resistance [10,38]. The conclusion that lipopolysaccharide-producing bacteria are enriched in obese humans is consistent with our findings. We observed that Desulfovibrionaceae was enriched in obese humans in our results. Lipopolysaccharide (LPS) causes obesity via an inflammation-dependent pathway [39]. LPS from members of the families Desulfovibrionaceae exhibits an endotoxin activity that is 1000-fold that of LPS from the family Bacteroideaceae [40].

We observed that the abundance of *Ruminococcus-1*, *Akkermansia*, *Lachnospiraceae_NK4A136**_group*, Christensenellaceae were decreased in the simple obesity group in our study. The latest research suggests that *Ruminococcus gnavus* was positively associated with percent body fat [18], inconsistent with our research. Contrary results may be related to different recruitment criteria. It has been demonstrated that *A. muciniphila* can alleviate diet-induced obesity by increasing energy excretion in the feces [41], indicating that *Akkermansia* insufficiency or deficiency may promote obesity by increasing food energy efficiency. Therefore, *Akkermansia* is regarded as a novel candidate to prevent or treat obesity. Weight-reduction interventions, such as dietary changes [42] and bariatric surgery [43], are often accompanied by partial restoration of microbial dysbiosis. Certain gut microbial strains have been shown to aid in weight loss in experimental models. For example, animal studies found that Lachnospiraceae administration alleviated obesity [44]. Further associated with several microbial functional pathways, such as butyrate-producing pathway (PWY-5022), amino acid biosynthesis and degradation pathways (HISDEG-PWY, DAPLYSINESYN-PWY), Lachnospiraceae bacterium 3 1 57FAA CT1 mediated the association of overweight/obesity with the Homeostatic Model Assessment of Insulin Resistance [45]. Goodrich et al. showed that *C. minuta*, a cultured member of Christensenellaceae, transplanted into germ-free mice reduced weight gain in 2014 [46]. Animal studies found that *C. minuta* DSM33407 protected against diet-induced obesity [47]. Mazier, Wilfrid et al. assessed the safety and tolerability of *C. minuta* DSM33407 in a phase 1 clinical trial [47]. In addition, *Alistipes onderdonkii*, *Alistipes finegoldii*, and *Alistipes shahii* were significantly higher in those individuals who showed improvements in the android-to-gynoid fat ratio within the fecal microbiota transplantation (FMT) group [48]. In a randomized, double-masked, placebo-controlled trial of adolescents with obesity, we observed a reduction in abdominal adiposity, increasing the relative abundance of Prevotellaceae in FMT recipient microbiomes by at least eight-fold postdosing [49]. The abundance of Erysipelotrichaceae was higher in the normal weight group in our study. It also was observed that the genus *Erysipelotrichaceae_UCG-003* was significantly more abundant in the gut microbiota of the healthy aging group versus non-healthy aging group [50].

The adenine and adenosine salvage pathway (PWY-6609), lactose and galactose degradation I (LACTOSECAT-PWY), superpathway of thiamin diphosphate biosynthesis II (PWY-6895), pyrimidine deoxyribonucleosides salvage (PWY-7199), and the L-histidine degradation I pathway (HISDEG-PWY) were clustered in obese patients. Adenine and adenosine salvage III are responsible for the conversion of the nucleotides to the nucleoside (adenosine) and free base (adenine) forms. All adenosine receptors have been reported to be involved in glucose homeostasis, inflammation, adipogenesis, insulin resistance, and thermogenesis, indicating that adenosine could participate in the process of obesity [51]. Histidine can be metabolized to imidazole propionate [52], which was found to be elevated in individuals with type 2 diabetes mellitus, impairing insulin signaling through activation of the p38γ–p62–mTORC1 pathway [53]. Lower histidine concentrations were observed in obese women than in nonobese women [54]. Our study showed that the L-histidine degradation I pathway was enriched in obese patients. In a Chinese supplementation study, obese women who received 12 weeks of supplemental histidine experienced decreases in body mass index, waist circumference, and body fat [55]. This finding indicates that dysbiosis of intestinal flora may play an important role in the pathogenesis of obesity by affecting amino acid bioavailability to the host [56].

The capability of microbial markers for the early prediction of obesity was comprehensively assessed in our study. The best-performing model achieved a high accuracy (AUC = 0.83) with 38 important features to distinguish simple obesity from normal weight. For a Chinese population, six biomarkers were identified to differentiate obese patients and healthy individuals through random forest classifiers. However, a limitation of the research is that the mediocre accuracy (AUC 0.68) was not tested in different populations [57]. With LOOS validation across multiple datasets, the important features could overcome technical and geographical discrepancies.

This is an individual participant data meta-analysis with standardized bioinformatics pipelines clarifying the differences in fecal bacterial diversity and communities in patients with simple obesity. We found that the microbiome composition in patients with simple obesity did differ significantly. At the phylum level, the Bacteroidetes/Firmicutes ratio was associated with the body mass index. We also identified a universal set of microbial markers to predict obesity. Importantly, we explored the relationship between the metabolic pathways of the intestinal flora and simple obesity. Despite including simple obesity without complications and comorbidities, excluding confounding by comorbidities or medications, no detailed adjustments for relevant confounders were performed. We also recognize the limitations of this study. The studies included in our analysis were based on 16S rRNA sequencing rather than shotgun metagenomic sequencing data.

## 5. Conclusions

Evaluation of universal gut microbiota biomarkers in obese individuals can be applied for the early prediction and potential gut microbiota targets for adjuvant treatments of obesity given that the gut microbiome plays an important role in the onset and progression of obesity [14]. Overall, our study identified universal biomarkers for obesity prediction and therapeutic targets. The RF model can help us choose the most suitable bacterial strains to utilize precision medicine, with greater benefits for obese patients.

## Figures and Tables

**Figure 1 nutrients-14-02993-f001:**
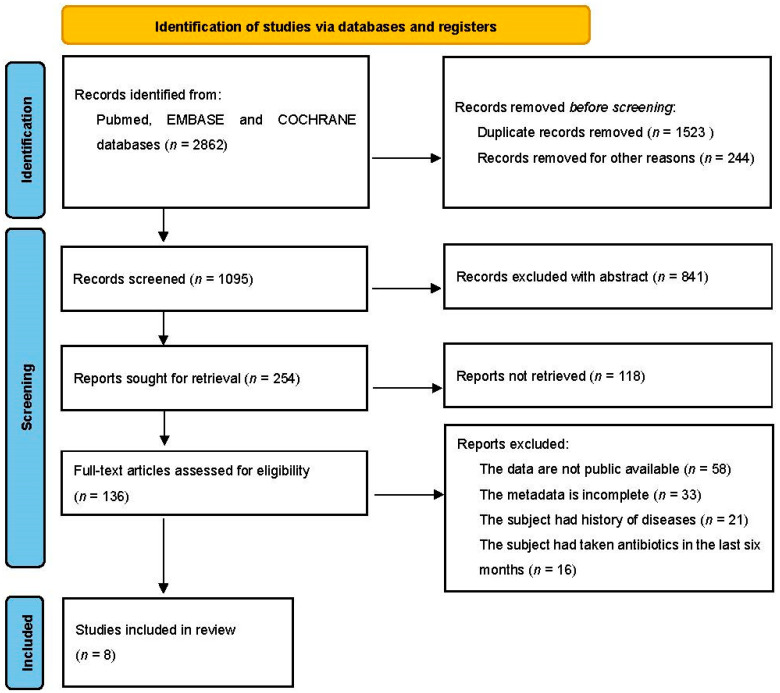
Description of the selection of the included studies following a PRISMA flow diagram.

**Figure 2 nutrients-14-02993-f002:**
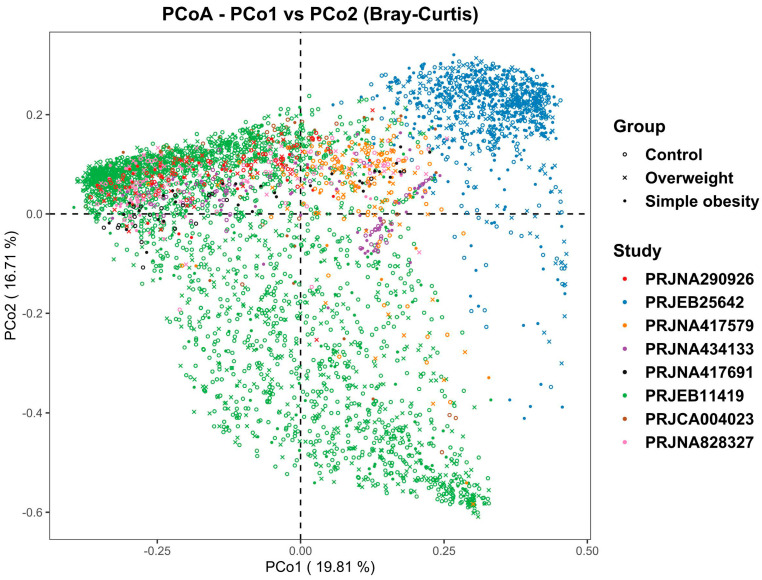
The principal coordinates analysis (PCoA) with Bray-Curtis distance based on genera. Each point in the diagram represents a sample. The shapes represent the control, overweight, and simple obesity groups, respectively. The colors represent the different studies.

**Figure 3 nutrients-14-02993-f003:**
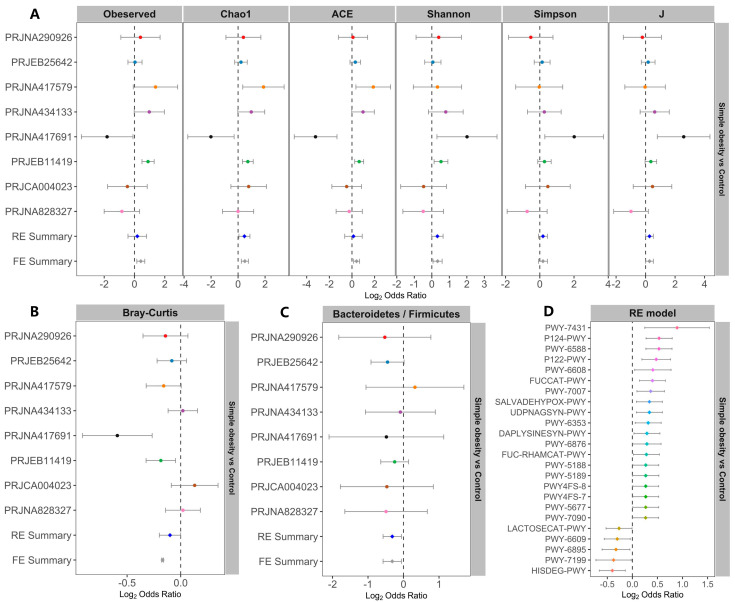
Comparison of bacterial alpha-diversity, beta-diversity, the Bacteroidetes/Firmicutes ratio, the pathway metrics between the simple obesity and control groups. Forest plots of (**A**) the alpha-diversity metrics, (**B**) Bray-Curtis distances, (**C**) the Bacteroidetes/Firmicutes ratio, and (**D**) the pathway metrics between the simple obesity and control groups. The error bars describe the 95% confidence intervals. A value less than 1.0 (the left of the dotted lines) indicates that the metric is higher in the simple obesity group than in the control group. The values larger than 1.0 (the right of the dotted lines) indicate that the metric is lower in the simple obesity group than in the control group. If the dotted line and the error bars did not cross, there is a significant difference between the simple obesity and control groups.

**Figure 4 nutrients-14-02993-f004:**
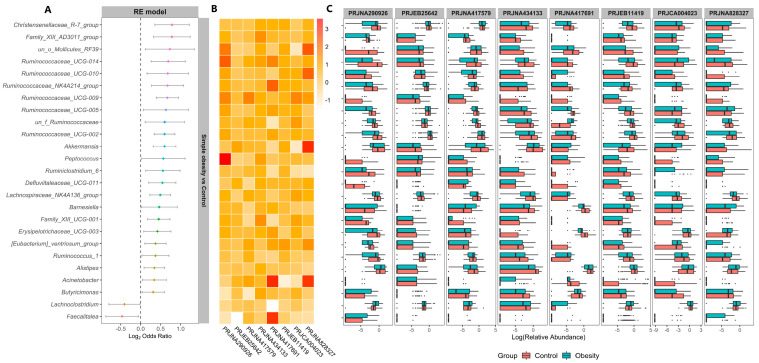
Discriminative taxa between the simple obesity and control group. (**A**) Forest plot, (**B**) fold changes of control to simple obesity, and (**C**) relative abundances of genera with significant ORs obtained by the random effects model analysis between the simple obesity and control groups. The color block values greater than 0 indicates that the genera were enriched in the control groups. Log transformation is applied to fold changes and relative abundance.

**Figure 5 nutrients-14-02993-f005:**
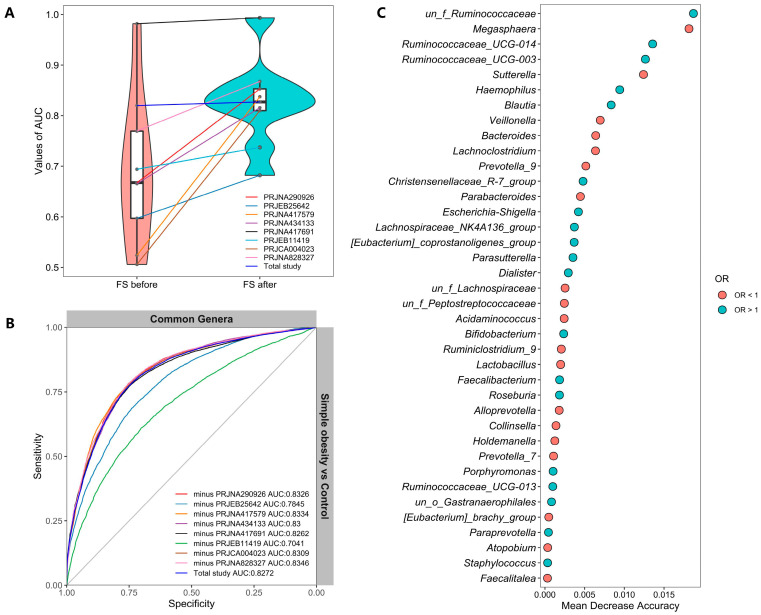
The performances of models to classify the simple obesity and control groups based on common genera. (**A**) Improvements in model performances were observed after feature selection (FS). (**B**) ROC of training data of the leave-one-out study models and training data pooled studies based on selected features. (**C**) Importance and ORs of features selected in the model based on pooled studies.

**Table 1 nutrients-14-02993-t001:** Characteristics of the datasets included in the fecal sample-based analysis.

Source	Year	Country	Control	Overweight	Obesity	DNA Extraction	Region	Sequencing Platform
PRJNA290926 [20]	2016	USA, Canada	44	48	33	PowerSoil-htp 96 Well Soil DNA isolation kit	V4	MiSeq
PRJEB25642 [23]	2018	India	228	259	425	QIAamp DNA Stool Mini Kit	V4	Ion torrent
PRJNA417579 [22]	2019	Columbia	87	57	20	QIAamp DNA Stool Mini Kit	V4	MiSeq
PRJNA434133 [24]	2019	UK	61	-	71	PSP Spin Stool DNA Plus Kit	V4	MiSeq
PRJNA417691 [21]	2019	Mexico	20	-	30	ZR Faecal DNA MiniPrep	V3	Ion torrent
PRJEB11419 [25]	2019	USA, UK	1697	739	240	-	V4	MiSeq, HiSeq
PRJCA004023 [13]	2021	China	37	-	37	QIAamp DNA Stool Mini Kit	V3-V4	MiSeq
PRJNA828327	2021	China	62	49	38	QIAamp Fast DNA Stool Mini Kit	V4	MiSeq

**Table 2 nutrients-14-02993-t002:** Importance, odd ration, confidence interval, and relative abundance of the eight genera selected for the RF model for simple obesity based on all samples.

Genera	Mean Decrease Accuracy	OR	CI_ub	CI_lb	*p*-Value	Abundance (%) in Control	Abundance (%) in Simple Obesity
*Christensenellaceae_R-7_group*	4.80 × 10^−^³	1.726	1.280	2.326	3.45 × 10^−4^	1.707 ± 0.03	1.098 ± 0.02
*Ruminococcaceae_NK4A214_group*	1.56 × 10^−^³	1.596	1.209	2.107	9.61 × 10^−4^	0.488 ± 0.01	0.437 ± 0.01
*Akkermansia*	1.05 × 10^−^³	1.514	1.242	1.845	4.11 × 10^−5^	2.169 ± 0.07	0.985 ± 0.04
*Ruminiclostridium_6*	3.85 × 10^−4^	1.471	1.092	1.982	1.11 × 10^−2^	0.375 ± 0.01	0.175 ± 0.01
*Barnesiella*	6.29 × 10^−4^	1.380	1.009	1.888	4.39 × 10^−2^	0.453 ± 0.01	0.237 ± 0.01
*Alistipes*	2.18 × 10^−^³	1.269	1.060	1.520	9.58 × 10^−^³	1.798 ± 0.02	1.148 ± 0.03
*Butyricimonas*	5.67 × 10^−4^	1.243	1.024	1.509	2.82 × 10^−2^	0.084 ± 0.002	0.063 ± 0.002
*Lachnoclostridium*	6.37 × 10^−^³	0.755	0.575	0.990	4.18 × 10^−2^	0.451 ± 0.01	0.516 ± 0.01

Note: CI_lb, confidence interval_lower bound. CI_ub, confidence interval_upper bound.

**Table 3 nutrients-14-02993-t003:** Importance, odd ration, confidence interval, and relative abundance of the four genera selected for the RF model for overweight based on all samples.

Genera	Mean Decrease Accuracy	OR	CI_ub	CI_lb	*p*-Value	Abundance (%) in Control	Abundance (%) in Overweight
*Succinivibrio*	5.33 × 10^−^³	0.656	0.509	0.846	1.13 × 10^−^³	1.525 ± 0.03	3.679 ± 0.12
*Christensenellaceae_R-7_group*	3.10 × 10^−^³	1.335	1.029	1.732	2.99 × 10^−2^	1.78 ± 3 × 10^−4^	1.356 ± 0.02
*Hydrogenoanaerobacterium*	6.62 × 10^−4^	1.928	1.535	2.423	1.71 × 10^−8^	0.007 ± 0.02	0.003 ± 2 × 10^−4^
*Methanobrevibacter*	2.67 × 10^−4^	1.358	1.132	1.629	9.94 × 10^−4^	0.227 ± 0.08	0.168 ± 0.01

## Data Availability

Not applicable.

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
