# Peer review of "Gut Microbiota Characteristics of People with Obesity by Meta-Analysis of Existing Datasets"

_nutrients, 2022, doi:10.3390/nu14142993_

Round 1

Reviewer 1 Report

This manuscript performed integrated analysis of normal weight, overweight, and obesity to identify obesity-associated microbial markers. It gives interested and useful information, however, there were something to be improve.

General comments:

1) This is not longitudinal study. Therefore, I think the authors should not use "change", "increase", and "decrease" for the microbial community composition.

2) The authors targeted to "simple obesity without complications and comorbidies" (L.100). However, in the Results and discussion, the authors express subjects as "patients with obesity" (e.g., L.257).  I think the authors should unify the expression for the subjects. 

3) The performed integrated analysis of normal weight, overweight, and obesity to identify obesity-associated microbial markers. It gives interested and useful information, however, there were something to be improve.

General comments:

1) This is not longitudinal study. Therefore, I think the authors should not use "change", "increase", and "decrease" for the microbial community composition.

2) The authors targeted to "simple obesity without complications and comorbidies" (L.100). However, in the Results and discussion, the authors express subjects as "patients with obesity" (e.g., L.257). I think the authors should unify the expression for the subjects. 

3)The descriptions about obese and overweight were mixed and it makes me difficult to understand the difference between them. 

Specific comments:

(L.200-201) "...ORs higher than ... " . This sentence should be improved. I think "between" is not suitable.

(L.221-225) This sentence is too long to understand.

(L. 289 and L.296) I cannot find "Figure 11B" and "Figure 11C". Please check the figure numbers throughout the manuscript.

(L.321) "The gut microbiota affects the energy balance of the host by regulating the genes related to fat absorption and storage". Would you please add the reference.

(L.384) "This is the first patient-level meta-analysis..." I think this expression should be revised.

(Figure 2) The color difference between PRJNA417691 and PRJEB11419 should be more noticeable.

(Figure 3) I think the results of PRJNA417691 is a little different from other studies. Would you please add the description and its reason.

Reviewer 2 Report

This manuscript is an interesting and well conducted meta-analysis identifying obesity-associated gut microbial markers.

The paper is overall well written, English use is good. However, some concerns and possible improvements have been identified. Specific comments are detailed below.

Abstract:

-Line 22. Please, specify that is a meta-analysis, otherwise it does not emerge from the abstract. Maybe “meta-analysis” should be added also in the title.

1. Introduction

-Line 72.Change “They” to: “The Authors”.

-Lines 79-80. Modify the sentence as follows: “..is that analysis by bioinformatics methods to identify specific genes and pathways was not performed”.

2. Materials and Methods

2.1. Database search and study selection

-Line 100. Change “is” to: “was”.

Table 1

Add in the Table the references to the 8 studies. I think this is an important information missing.

            Source PRJNA434133. The “DNA extraction” column indicates a kit that is for RNA extraction. I imagine this is an error.

3. Results

3.1 Microbiome profile differences among the obesity, overweight and control groups

-Lines 192-194. This is the most critical point. This sentence is unclear. For what I understand, considering that Figure 2 clearly shows a clustering according to the different studies, and not to the different groups (obesity, overweight and control), a new analysis has been performed, in the attempt to minimize the effect of the studies and amplify differences due to the groups. If it is so, I think that the data of this new analysis should be presented as an additional Figure. Otherwise, the Authors should clarify this aspect, as the data shown in Figure 2 weaken somehow all the Results.

-Lines 221-226. This part on metabolic pathways should have more emphasis. As it is, it “gets lost” among the other results. I suggest to amplify this part, recalling the full names of the pathways reported in the Abstract and inserting full lenght names of all the acronyms.

-Line 245. Correct “genera” to: “genus”.

Tables 2 and 3.

Explain in a footnote what “ub” and “lb” are.

4. Discussion

-Line 315. Remove “while”.

-Line 319. Unclear sentence.

-Line 346, Put “C. minuta” in italic.
